# Building a Parkinson-Network–Experiences from Germany

**DOI:** 10.3390/jcm9092743

**Published:** 2020-08-25

**Authors:** Marlena van Munster, Lars Tönges, Kai F. Loewenbrück, Tobias Warnecke, Carsten Eggers

**Affiliations:** 1Department of Neurology, University Hospital Marburg, 35033 Marburg, Germany; munster@med.uni-marburg.de; 2Department of Neurology, St. Josef-Hospital, Ruhr-University Bochum, 44801 Bochum, Germany; lars.toenges@rub.de; 3Neurodegeneration Research, Center for Protein Diagnostics (ProDi), Ruhr University, 44801 Bochum, Germany; 4Department of Neurology, University Hospital Dresden, 01307 Dresden, Germany; kai.loewenbrueck@uniklinikum-dresden.de; 5German Center for Neurodegenerative Diseases (DZNE) Dresden, 01307 Dresden, Germany; 6Department of Neurology, University Hospital of Münster, 48149 Münster, Germany; tobias.warnecke@ukmuenster.de

**Keywords:** Parkinson’s disease, integrated care, multi-disciplinary care, network, network design

## Abstract

Parkinson’s disease is a complex neurodegenerative disease that can be best treated with a multi-disciplinary care approach. Building care networks has been shown as a useful tool to facilitate the integration of care services and improve outcomes for patients and care providers. However, experiences and practices relating to building a network are very limited in the field of Parkinson’s disease. This paper portrays existing Parkinson networks in Germany. With the help of a standardized template, description of networks and their building-blocks, so-called modules, were collected from all over Germany. Modules were rated in terms of their expected benefit and the required effort when implementing them, with the help of an expert survey. The rating showed that some modules were perceived as more important than others, but all modules were recognized as beneficial for patients and care providers. Overall, the German experience shows that building a Parkinson network facilitates the integration of care and provides a benefit to all stakeholders involved.

## 1. Introduction

Neurological disorders are the leading cause of disability in the world, among which Parkinson’s disease (PD) is the fastest-growing [1]. In Germany, PD is the second most common neurodegenerative disease and an increasing prevalence is observable [2]. According to data from the Federal Statistical Office, were 11,190 deaths related to the primary Parkinson’s syndrome in 2018 [3]. The Germany society is aging and age-related diseases like PD will continue to increase [4]. Currently, it is assumed that there are around 420,000 PD patients in Germany [5]. PD is characterized by a progressive course, which includes complex and often changing motor and non-motor symptoms, such as fatigue, depression and autonomic dysfunction [6]. Patients require the long-term, comprehensive provision of care in outpatient and stationary settings consisting of medical management and non-pharmacological measures, including physiotherapy or occupational therapy [4]. Additionally, relatives require support, such as consultations and guidance [6]. In Germany, PD treatment is mostly limited to symptomatic therapy [7]. Treatment plans must be adapted continuously to the patients’ disease progression and symptoms [7]. Thus, PD requires accessible, long-term, multi-disciplinary, patient-centered treatment and healthcare support [8,9,10].

Integrating care services has been shown as an effective tool to meet these requirements [11,12] and as a method to enhance patients’ quality of life and access to health care [11,13,14]. Integration can be achieved in various ways. For example, in 2018 a local initiative in Germany fostered the communication between several PD care providers which significantly improved the quality of life in PD patients compared to the standard neurological practice [11]. Overall, the establishment of Parkinson networks has been shown as a useful tool to achieve the integration of care services [12,15]. Parkinson networks can be understood as a merger of several units that share and exchange data, as well as resources among each other [16]. They often aim to improve patient care across different sectors, avoid unnecessary hospitalizations and reduce costs. However, these networks can also be used to enable faster and more efficient PD diagnosis, optimize patient treatment and create better working conditions for healthcare providers [10,16].

Numerous studies have developed frameworks which set out important components of an integrated care network [9,15,17,18]. One framework that has also been used to build successful integrated networks is the development model for integrated care (DMIC) [15,19]. The model defines nine generic elements, which are essential across settings and patient groups in order to establish a functioning integrated care network, namely: quality care, performance management, inter-professional teamwork, delivery system, roles and tasks, client-centeredness, commitment, transparent entrepreneurship and result-focused learning. PD patients require care services that go beyond these generic themes and need to be considered when building a network, which is why specific recommendations have been made for organizing integrated PD care [20,21].

In Germany, there are no well-established multi-disciplinary care structures for neurological diseases in general. However, when it comes to PD, care concepts have developed dynamically within the past years and today several integrated care networks for PD patients (usually also including patients with atypical Parkinsonian disorders) exist in Germany, for example, in Münster, Dresden and Marburg. Additionally, the German Society for Parkinson and Movement Disorder (DPG) established a working group entitled ‘Networks and Care’ to facilitate knowledge exchange and collaborative learning [7]. On an international level, a task force was founded in 2018 by the International Parkinson and Movement Disorder Society with the goal to establish best-practice-models, identify outcome parameters for future studies and give recommendations for the development of care networks [11].

However, there is still room for improvement. A survey of German PD patients in 2018 showed that the first point of contact usually is in the outpatient sector with resident neurologists being the main care provider to the patients (85.2%, *n* = 1179) [13]. Most of the PD patients consult a resident neurologists or psychiatrists (90.8%; *n* = 1156), a hospital (21.2%; *n* = 255) and a specialized Parkinson clinic (17.7%; *n* = 225) at least once per year [13]. Experts at hospitals and clinics are important stakeholders in the statutory care sector since they provide advanced treatment methods (i.e., deep brain stimulation (DBS)) [13]. Currently, there is no regular mechanisms to interlink statutory and outpatient care. This can lead a non-coordinated approach to diagnostics and therapy [13]. Furthermore, healthcare in Germany is organized on a regional level and Parkinson networks heavily rely on local initiatives [13,22]. At the moment, neither all PD patients have access to Parkinson network, nor it is guaranteed that all PD patients within the reach of a network are referred to it by their primary care provider [7]. As a result, there is a call for the establishment of national standards at the health policy level and the systematic recording of initiatives to improve cross-sector care to enhance the level of integration and further improve care for PD patients [11].

Today, the establishment of a Parkinson network, which interlinks important care providers, such as resident neurologists and experts at a hospital, is time and resource consuming. The networks have to come up with own strategies to connect with local care providers and patients and currently, there are only little practice-based examples how this could be achieved [9]. This paper presents an overview of interventions (so-called ‘modules’) which have been used to build these networks and, for example, helped to ensure that resident neurologists indeed refer their PD patients to the dedicated events of a network.

Overall, this paper has a two-folded aim: first, it aims to present all Parkinson networks and modules, which are present in Germany. Second, it aims to highlight the contribution of each module to the establishment of an integrating care network and provide an expert judgement on their costs and benefits. Finally, recommendations for building a network based on German experiences will be given.

## 2. Methods

In order to fulfill the first purpose of the investigation (to present an overview of all Parkinson networks and their modules) information about existing modules and networks was collected by using a Microsoft Word template. The template was developed by the DPG working group ‘Networks and Care’ in which the authors participate to ensure a structured collection process. The template asked about the goal, strategic focus, participating parties, contractual cooperation partners, the legal framework, funding mechanisms, the structure and frequency of cooperation and externals. It was used to collect information from October 2019 to December 2019 about networks and modules among experts from all over Germany. To ensure that all existing networks and modules have been compiled, the heads of each Parkinson network were identified via the DPG registry and invited by mail to contribute to the study. The authors had access to the DPG registry since they are members of this association as well. Additionally, an invitation in the form of an electronic newsletter was sent to the German PD expert community to participate in the study. Again, these experts were identified with the help of the DPG registry and included doctors and scientists focusing on care supply for PD patients. In both cases, the template was attached to the invitation and participants were asked to describe a Parkinson network in which they participate and/or as many network modules as they wish by filling out the template. The participants were instructed to fill out one template for each module or network and sent them back to the investigators by mail. In order to provide an overview of the costs and benefits of each module, an anonymous web-based survey was conducted in 2020 from June 29th to July 11th among the participants from the template collection. They were invited to participate in the survey by email, which also included the link to access the survey. The structure and content of the online survey can be accessed in the Appendix A. The experts rated each identified module on a five-point Likert-scale (1 = none, 5 = very much) according to the additional workload caused by its implementation, the expertise and amount resources required and the expected benefit for patients and specialists. Using a five-point Likert-scale as tool to retrieve expert opinions is a widely used method in integrated care research [23,24,25]. The rating criteria have been selected based on the recommendations for the organization of multi-disciplinary PD care teams [20]. The survey was analyzed by using Microsoft Excel to calculate the average rating points, as well as the standard deviation, for each category and each module. The results have been converted into a visual scheme to provide a better overview. Additionally, the modules have been grouped into the nine categories of the DMIC to facilitate a better overview of their part within an integrated care network. For the purpose of this study, PD specific criteria for care have been added to the model and the term ‘client-centeredness’ has been replaced by the term ‘patient-centeredness’ [20]. An overview of the categories can be derived from Figure 1.

## 3. Results

Seven Parkinson networks (*n* = 7), compromising a total of twenty-five modules (*n* = 25) were identified. Twenty-one experts (*n* = 21) contributed templates about networks and modules and 81% (*n* = 17) participated in the online survey. 

### 3.1. Parkinson-Networks in Germany

As mentioned before, the modules were collected from seven Parkinson networks across Germany. All templates that were handed in and describe a network can be accessed in the Appendix A. Their location is displayed in Figure 2. What becomes present is that not all German regions have implemented a Parkinson network. The networks are presented briefly in the following.

#### 3.1.1. Hamburg (Satellite Network Hamburg)

The Satellite Network in Hamburg (North Germany) was initiated in 2011 by Dr. med Wellach and Dr. med. Becker. It aims to improve the link between in- and outpatient care in PD during all phases of illness and optimize interface management in order to provide interdisciplinary, inter-professional and cross-sectoral care. Strategic priorities are interlinking in- and out-patient care through cross-sectoral medical activity, increasing the use of Parkinson nurses and Parkinson assistant (PASS), networking with other professionals and using technology (e.g., outpatient video-supported therapy). Currently, various stakeholders participate in the network, such as neurologists, hospitals, outpatient care facilities, therapists and patient associations. One partner is involved via a contract-based cooperation and another partner (Association for Quality Development in Neurology and Psychiatry—QUANUP e.V.) is involved via the membership of other network partners. However, most partners are involved through voluntary based cooperation. The network is financed through earnings from lectures and trainings and with the help of individual funding projects. External funding is currently not available. Interaction between partners takes place in terms of content-exchange about topics such as direct patient care and professional education. The partners communicate with the help of a communication platform, via written instructions in the German health insurance Safenet and via other channels, such as phone calls. From time to time, joint publications, advanced training events and trainings for Parkinson nurses and PASS are organized. The frequency of interaction is need-oriented and varies.

#### 3.1.2. Lutherstadt (Parkinson Network Central Germany)

The network in central Germany was initiated in 2009 by Dr. Feige. It aims to connect in- and out-patient care for PD patients through the establishment of a PD specialist clinic as a statutory care provider. Additionally, the clinic functions as partner and advisor for resident neurologists, self-help groups, affected persons and relatives within the region. Additionally, three other clinics and a health professional association are involved via contract-based cooperation. Overall, the German regulations for interactions between doctors and patients form the legal framework for joint actions. Besides the usual costs, financial resources are only required for national specialist and patient events. These events are supported by the pharmaceutical industry and the clinic. Overall, the network facilitates equal cooperation among the partners. A patient event is organized every two years. Lectures are held annually in the clinic and external lectures are organized several times a year. Externally, the network is presented via annually updated flyers and a seal of quality (certificate) from the German Parkinson Association (DPV e.V.), which is renewed every three years.

#### 3.1.3. Jena

This network was founded in 2019 by the Center for Movement Disorders at the Jena University Hospital (Jena, Thuringia, Germany). The recently founded network aims to connect in- and out-patient care for PD patients. Specifically, the focus rests on optimizing case selection, providing training of non-specialized neurologists, general practitioners and therapists and empower patients. The Center for Movement Disorders coordinates the network; other partners are resident neurologists, practices for physiotherapy, occupational therapy, speech therapy and self-help groups. Currently, no contracts or a legal framework exists. Financial resources come from the hospitals’ budget. Partners collaborate in various way. For example, by organizing professional trainings or patient events. Additionally, the network uses telemedicine to enhance care provision. Partners currently communicate via phone and mail, but a communication platform is planned. In order to represent the network externally, a homepage is under construction.

#### 3.1.4. Düsseldorf

The Parkinson network in Düsseldorf was initiated around 2010 at the Neurological University Clinic Düsseldorf (Prof. Dr. Schnitzler) and is currently coordinated by Prof. Dr. Groiss. The network aims to acquire patients for DBS by providing consultations to resident neurologists and certified training events in the Center for Movement Disorders and Neuromodulation. The consultations were held initially every three months in practices of resident neurologists together with a senior doctor from the clinic. Currently, these sessions are held twice per year. A legal framework or financing strategy is not established since partners cooperate on a voluntary base. Partners collaborate in the form of collegial joint consultation hours, which are held by appointment. Externally, the network is presented online.

#### 3.1.5. Münster and Osnabrück (Parkinson Network Münsterland+; PNM+)

The PNM+ was initiated in 2017 by the Department of Neurology at the University Hospital of Münster in cooperation with an industry partner and aims to optimize the supply chain of care for PD patients and their families. Strategic priorities are to enhance the coordination of care for patients with the help of exchange and networking, the creation of interdisciplinary care team with pooled expertise, as well as, sharing experiences and building knowledge. The network includes clinics, general practitioners, specialists, various therapists, pharmacists, rehabilitation clinics, patients and their relatives, self-help groups, Parkinson nurses, PASS and industry partners. Partners participate in the network via membership and sign a cooperation agreement, which includes working principles, rights and obligations. Additionally, contract cooperation’s with other stakeholders are established. In the initial phase, the network was funded with the help of cooperation funds from an industry partner and equity from the hospital. Other financing mechanisms are planned for the future, such as donations and sponsorship. Partners meet in quarterly plenary meetings. Additionally, working groups (WG’s) have been established which take over internal coordination and special tasks. A steering committee manages the overall network. Partners communicate via a platform and present themselves externally on a homepage.

#### 3.1.6. East Saxony (Parkinson Network East Saxony; PANOS)

The PANOS network was initiated in 2018 by Dr. Wolz (Meissen Elbland Clinic), Dr. Löwenbrück (University Hospital Carl Gustav Carus Dresden) and Dr. Themann (Clinic at Tharandter Wald Hetzdorf). It aims to improve patient’s quality of life with the help of a regional inter-sectoral structured treatment path, where care is delivered timely and equal to all patients, regardless of where they live and other socio-demographic factors. The network has various strategic priorities centered around implementation goals, such as the establishment of three regional specialized outpatient care facilities for PD. Other priorities are infrastructure-related, such as the establishment of a lifelong personal inter-sectoral case management (‘Parkinson’s Guide’), active network management or the establishment of a standardized patient school. The network follows a sequential implementation strategy: In a first phase, it will compromise the three previously mentioned clinics, general practitioners, specialists, community-based neurologists, scientific institutes, evaluation partners (health insurances and research departments), associations and professional organizations. In a second phase, the integration of further relevant healthcare professions is planned, e.g., various therapists and pharmacists. The network has a financing strategy: all partners are cooperating on the bases of cooperation agreements within a publicly funded health services research project, except of the health insurance companies since they have their own budget. All partners are involved in the network management and needs are frequently assessed via workshops. The partners work in topic-related inter-sectoral WG´s on the overall goals. The external representation is currently under construction.

#### 3.1.7. Marburg (Parkinson Network Alliance Marburg; PANAMA)

The PANAMA network was initiated in 2016 by the Department of Neurology at the Marburg University Hospital. The network aims to pool work related to PD, train and educate laypersons and specialists, and connect regional supply partners, for example, by organizing events such as the Parkinson Summer Festival. Strategic priorities are an optimized collaboration with referrers, treating complex patients in the hospital and enable continuous development of care options for them. Additional priorities are the identification of patients for invasive therapies, such as DBS, and fostering scientific research in the field of PD. Currently, several care providers from the region participate in this network, such as clinics, therapists, rehabilitation facilities and resident doctors. The partners cooperate on a voluntary base. Thus, no legal framework exists. They work together by participating in joint training events (i.e., at the Parkinson Board), consulting local partners in order to identify patients for DBS and creating possibilities for unstructured exchange (i.e., at the Parkinson Academy). Until now, no financing structure is established. In the past, the network was funded by sponsoring for lectures, benefits from providing further education and single funding projects. A systematic communication channel has not been established yet and partners correspond via e-mail or phone calls. The network is presented in the media, at lectures and online.

### 3.2. Network Modules

Next, the network modules are presented which have been collected using the template among PD experts working in one of the seven previously outlined Parkinson networks. All templates that were handed in and described a network module can be accessed in the Appendix A. The modules are organized in an alphabetical order.

#### 3.2.1. Deep Brain Stimulation Consultation

In patients with advanced PD, intensified therapies (e.g., DBS) are often required. These therapy options require a close interlink of statutory and outpatient care [7]. However, as mentioned previously this is often not the case in Germany. This module aims to advise patients on DBS. The focus rests on providing individual advice and risk assessment regarding the indication for DBS. In this module, a resident neurologist (outpatient care) and a specialist from a university hospital (statutory care) participate. The cooperation is informal. Thus, no legal framework, contract or funding is required. Joint consultations with the practice owner, the neurosurgeon and the patient are held, where the specialist advises the patient while taking into account the individual course. Appointments are made via telephone agreement between the practice and the neurologists’ secretariat. When integrating care services, this module forms a part of the delivery system and can be used to improve patient-centeredness.

#### 3.2.2. Evaluation

Evaluating the network helps to investigate its effectiveness (as a whole or for single projects), measured against specific and transparent criteria. Strategically, the definition of a target population, the project, which will be evaluated, and appropriate parameters are vital points. Besides the steering team, a designated WG should be involved in this module. Additionally, an independent external partner (e.g., a research institute) is required to perform the evaluation. A contract needs to be set up with this partner and financial resources need to be reserved. Based on previous experiences, financial resources between 20.000€ and 70.000€ are required. The steering committee and a designated WG will establish the target, project or process to be evaluated and develop evaluation criteria in cooperation with the external partner. The partner typically prepares the evaluation based on questionnaires filled out by all network partners. Further, the results may be used by the public relations team. During the assessment, frequent meetings are required. Externally, the results can be distributed within the region, in journals or via press releases. Evaluating the network is a fundamental part of managing its performance, improve quality of care and enable result-focused learning.

#### 3.2.3. Financing Strategies

Financing strategies are set up to secure a long-term persistence of the network and finance projects, speakers, trainings or external partners. When deciding on a strategy, some fundamental decisions have to be made (e.g., should there be a membership fee for network partners?). Additionally, it may be attractive to use public and/or health insurance funding. All partner should be involved in this module. A legal framework is usually required, depending on the financing strategy (e.g., when industry partners provide a sponsorship). Additionally, costs might arise from travelling. The steering committee has to analyze different strategies, present them to the network partners and discuss which paths should be followed and in which order. When state funding is used, a designated WG should be founded to manage communication with the national funding department. The frequency of meetings depends on the strategy. Generally, financing strategies may differ depending on the regional context, the size of the network, the partners which are involved, and the overall purpose and it has been recommended that these factors should be taken into account when establishing a strategy [14]. Externally, transparency vis-à-vis partners online and, if necessary, towards the public has to be created. Financing strategies fundamentally contribute to the delivery system within the network and the quality of care.

#### 3.2.4. Homepage

The network homepage helps to attract potential partners within the region and informs patients, the society, politics and other networks. A website strengthens the commitment of the partners and demonstrates professionalism. Therefore, the focus should rest on the creation and implementation of a sound concept with the help of a web designer. The steering committee, a designated WG and a web designer may participate in this task. If the web designer is hired, a contract and financial resources will be required. The costs will depend on the homepage design. Cooperation will consist of the steering team discussing ideas with the web designer and continuous implementation and support between all involved parties. Depending on the content, coordination with individual network partners may be necessary which can be time-consuming, as logos and consent must be obtained. The collaboration is performed continuously until the homepage is going live. Afterwards, an update mode can be established, and tasks of the steering team can be delegated to the WG. The homepage is an incremental part of the networks’ external representation. Thus, attention can be drawn to the homepage via flyers, presentations, lectures or press releases. Overall, the homepage signals transparent entrepreneurship and improves the delivery system.

#### 3.2.5. Multi-Disciplinary Plenary Meetings

Plenary meetings make it possible for the partners to get to know each other. Additionally, these meetings provide a platform for regular exchange, voting about results, making suggestions to other project groups and collecting new ideas. Thus, these meetings can help to build trust and documenting progress towards overall goals. Additionally, these meetings may be used to enhance the motivation of general practitioners and resident neurologists to refer their PD patients to the dedicated events of a network. While legal frameworks are usually not required, funding may be needed for speakers, room rental, catering and project management (if not performed within the network). When implementing this module, all partners of the network should be involved. During the meetings, it may be useful to have representatives for each project team, a moderator and a record-taker involved. The steering team of the network might create an agenda for each meeting, collect proposals for input among partners, invite speakers and schedule future meetings. In practice, these meetings are performed every three months. However, project teams may meet more frequently. This module may be used for externally presenting the network, for example, by releasing a project report on completed topics. In general, this module helps to define clear roles and tasks within the network, foster inter-professional teamwork and result-focused learning.

#### 3.2.6. Multimodal Complex Treatment

The Multimodal Complex Treatment is an in-patient treatment program geared towards more advanced patients and very complicated cases. The treatment compromises on average, fourteen days of in-patient care with a multimodal assessment at the beginning and end [26,27]. Patients receive at least 7.5 h of therapy per week by various therapists, specifically tailored to their needs. The aim of this intervention is to improve motor and non-motor function resulting in enhanced quality of life in patients with PD. Furthermore, patient safety and satisfaction can be increased as well as the enhancement of loyalty towards the care center. In Germany, this intervention is reimbursed by health insurance and an additional legal framework is not required. During this time, patient appointments have to be scheduled and weekly team meetings within the interdisciplinary therapy team are required.

#### 3.2.7. Online Communication Platform

The aim of this platform is to enable easy communication between all network partners. It can be used to plan joint meetings, exchange and collect document or discuss certain projects. Additionally, the platform can provide an overview of potential contact persons for specific supply questions. All network partners should participate there. Establishing a platform does not require a legal framework. However, financial resources might be needed. If existing tools are used, the costs are usually around 80 Euro per month for renting the platform. One has to make sure that all partners have access to the platform and can participate in discussions or add contributions. Possibly, protected folders for the WG´s and the steering committee have to be created. The platform should be used continuously. When it comes to integrating care, the platform can help to organize inter-professional teamwork and serve as a tool for managing performance.

#### 3.2.8. Outpatient Video-Supported Therapy

Video therapy enables anti-Parkinson therapy in an outpatient setting. Short video sequences of the patient are recorded and evaluated by a neurologist. Direct telephone contact with the patient is essential. This module can be established between a resident neurologist, a neurological clinic, a specialized company and different health insurance companies. A cooperation agreement needs to be established between the participating parties. In Germany, the service provision within this module can be reimbursed by a health insurance company. There are three possible scenarios for interaction between the patient and the neurologist (practice doctor evaluates patient videos alone; practice doctor asks for support from the clinic; clinic evaluates videos apart). Within the German reimbursement scheme, this service is once per year possible. Externally, this service should be presented. This therapy method facilitates the use of technology, helps to enhance patient-centeredness and addresses the delivery system within the network.

#### 3.2.9. Parkinson Academy

The Parkinson Academy provides practical training of care specialists, such as doctors, nurses and therapists. Strategically, this module can be used to expand professional knowledge. Doctors and Parkinson nurses from a university hospital, settled neurologists and clinical neurologists can participate in this module. Depending on the topic, members of other specialist groups such therapists may participate as well. The training involves a maximum number of 15 people and includes practical sessions, a concise theoretical update on current news, a keynote speech on the main topic and bedside teaching with relevance to the Parkinson network. An overall legal framework is not required. Funding will be required but may be retrieved from sponsorship. The content of the academy needs to be organized and participants need to be invited. Based on past experiences, registration should be mandatory. Overall, this module can foster result-focused learning within the network.

#### 3.2.10. Parkinson Assistant (PASS) Training

In Germany, a specialized training for PD care provider exists. The aim of this training is to bring qualified, Parkinson-experienced medical assistants forward with the help of nationwide curricular trainings, consisting of basic and advanced courses. Medical assistants sponsored by pharmaceuticals or self-payers (practices) are participating in this module. The legal framework consists of training- and (if applicable) sponsorship-contract. Companies usually sponsor the training for a certain number of participants. Practices pay per participating medical assistant. The basic course is held on two weekends and the advanced course is held on a Saturday about six months later. This training brings new professionals into an integrated care network and improves the delivery system.

#### 3.2.11. Parkinson Board

The Parkinson Board provides an opportunity for doctors to discuss complex patient cases. The focus rests on professional exchange and strengthening team communication but also the interaction with referrers. PD experts within a university clinic and resident neurologist participate in this module. In order to implement the module, legal framework or funding mechanisms are not required. The consultations take place in a university clinic with a specialized PD unit, after a patient was registered via mail or phone by a resident neurologist. Next, discussion of causalities takes place on the Parkinson Board, either in the presence of the neurologist or in absentia on the interdisciplinary board. In the end, specific written diagnostic and/or therapeutic recommendations are submitted to the neurologist. Meetings can be organized either by appointment or take place in a weekly team meeting. This module can contribute to the overall integrated network by improving inter-professional teamwork and result-focused learning.

#### 3.2.12. Parkinson Nurse

The Parkinson nurse is a specialized nurse who enables competent care and advice for Parkinson’s patients. When implementing the module priorities rest on qualifying the nurse, increasing employee commitment and enabling qualified care. Regular nurses at university hospitals can participate in this module. Financial resources are required for training. Afterwards, nursing staff with additional qualifications can be assigned to the appropriate ward area to care for PD patients. Similar to the PASS training, this module brings new professionals into the network and improves the delivery system.

#### 3.2.13. Parkinson Summer Festival

This module provides an opportunity for informal exchange between patients, relatives and care providers. Additionally, this event may help to recruit study participants and increase patient retention. The summer festival may include different activities, lectures or information points for patients within a care facility. Time is required for planning the event and inviting patients and partners. However, partners participate voluntarily. Thus, a legal framework is not required. Usually, the event can be financed with sponsoring and donations. This module can be used to improve patient’s commitment and engagement in the integrated care network.

#### 3.2.14. Parkinson Symposium

The Parkinson Symposium offers scientific training from doctors for doctors about relevant topics to PD. Typically, such events can be funded via sponsorship and partners participate voluntarily so no additional resources will be required. However, before the event participants need to be invited, registrations have to be tracked and an agenda has to be drafted. In terms of integrating care, this module can improve inter-professional teamwork and provide a platform for result-focused learning.

#### 3.2.15. Parkinson’s Info-Cafe

The café enables informal exchange among patients, relatives and Parkinson nurses. The strategic focus rests on giving advice on nursing aspects of the disease and creating patient loyalty towards the network. A location, coffee and cake free of charge need to be organized. Additionally, the program may include a fixed item (e.g., reading, presentation of a new study, presentation of a new device). Besides these resources, no additional resources are required since financial support can be derived from participant donations. Participants can be invited via flyers and local press releases. Requiring registration is recommended. When integrating care services this module can help to strengthen commitment towards the network and improve patient-centeredness and engagement.

#### 3.2.16. Quickcards

Quickcards are a tool to promote interface communication within the network. They enable targeted referrals with additional information and a standardized exchange among care providers. Ultimately, this facilitates targeted and evidence-based therapy. Quickcards can be implemented for all interfaces and relevant topics. In order to implement them, an assigned project team, WG’s, a printing company and, if necessary, a graphic agency which supports their creation should be involved. Contracts are needed for the cooperation with the printing company and the graphic agency. Financial resources will be needed for printing. The costs depend on the design but usually do not exceed 100€ per card. The WG has to meet frequently when setting up the card. Additional communication might be required via the online communication platform, by mail or via conference calls. Later, annual meetings may be appropriate. These WG’s efforts, which interfaces are targeted, and results can be communicated externally (e.g., at congresses). In terms of integrating care, the cards contribute to the networks´ delivery-system and foster inter-professional teamwork.

#### 3.2.17. Regional Demand Analysis and Target Formulation

Analyzing regional demand may help to detect current issues, such as a lack of interface collaboration or barriers for patients to enter the network (i.e., finding reasons for why resident neurologist/general practitioners are not referring their patients to the network). From there, need-based goals can be derived. For example, if there is a regional lack of communication among care providers, a communication platform may be established. Strategically, actions are focusing on joint, multi-disciplinary, as well as, region-specific determination of needs and objectives. Prioritizing goals, according to the likelihood of realization and the time required for fulfilment, may help to achieve this aim. When implementing the module, no legal and financial resources are needed. All partners of the network should be involved in this step. Later, the steering team may formulate goals, prioritize actions and discuss these during regular meetings with all involved partners. Collaborating can take place in various forms, such as a moderated workshop or a written query and may be presented externally in the way of a framework, flyer or preamble. It might be practical to perform the actual demand-analysis during the first meeting with all partners. However, target formulation and achievement as well as the integration of needs should be performed continuously. Overall, this module can help to enhance stakeholder’s commitment and engagement, define clear roles and tasks and improve patient-centeredness.

#### 3.2.18. Regional Public Relations

This module aims to achieve and increase awareness of PD and de-stigmatize patients through education. Implementing public relations can help to become notified by politics, possible sponsors, and potential patients. Strategically, finding the right contact persons online and creating information material is essential. All partners and a steering team should be involved in these tasks. Neither legal frameworks nor financial resources are required. Cooperation among partners should start during the first plenary meetings: relevant contact persons from the relevant area should be retrieved among network partners and listed. A ‘public relation’ WG should be established, including a ‘network spokesperson’. The WG should continuously promote the network (e.g., in urban and regional networks) and organize publicity events. Collaboration among the WG’s is ongoing. Further, larger activities may need to be prepared during frequent meetings. No additional external representation is required. Working on public relations signals transparent entrepreneurship and may help to enhance the patient-centeredness of the network.

#### 3.2.19. Regional Utilities Atlas

The atlas aims to provide an overview of regional care providers with expertise in PD. The strategic priorities are to compile all relevant care providers in the region, including their expertise and to map the utilities on a network card. Additionally, it is crucial to have the right contact person on the map. From a supply perspective, the atlas can be used to identify gaps in the regional healthcare supply. When setting up the atlas, a designated WG and all members of the network should be involved. A legal framework is not required but it might be necessary to collect approval to store and map contact details. Financial resources depend on the presentation of the atlas. The partners need to agree on definitions of the Parkinson-specific aspects of care, which shall be mapped. Relevant information (e.g., contact details) needs to be retrieved in a written or digital form from network partners and regular confirmation or adjustment of these details needs to be provided from each partner. The frequency of this ‘check-up’ can be performed annually. The atlas can be displayed online to promote the network externally. However, before this can be done a consultation with the legal and data protection department is necessary. Otherwise, the atlas may be used internally. In terms of integrating care, the atlas contributes to the networks’ delivery systems and reflects transparent entrepreneurship.

#### 3.2.20. Rehab

This module compromises visits from an expert working in the statutory care sector (usually at a university hospital) at another hospital, which offers a rehabilitation program (outpatient care). During this visit, the expert educates the team about advanced treatment options (i.e., DBS) and gives advice on specific patient cases. The aim is to expand competence regarding advanced treatment options in external rehabilitation clinics and enhance patient safety and satisfaction. Similar to the DBS consultation, this module helps to connect the statutory and outpatient care sector. These visits are based on voluntary participation. Thus, legal or financial resources are not required. Participants and the referee have to plan in time for participating in the training and the external partner may invite specific patients.

#### 3.2.21. Standardized Treatment Path

The treatment path aims to deliver efficient, timely and equal care supply for all PD patients, regardless of where they live and other socio-demographic factors. Strategically the focus at the beginning is to develop a consensus-based inter-sectoral treatment path with the involvement of the regional outpatient sector and under consideration of (inter-)national recommendations. Tasks need to be defined and divided, minimum treatment requirements need to be defined and a regular patient monitoring schedule has to be established (currently ever three months). Additionally, personnel and technical implementation have to be organized and the network should grow within a controlled process. Finally, structures, processes and results need to evaluate. A cooperation agreement between participating partners, as well as financial resources, will be required. Partner collaborate based on needs and obligations within the agreement. Externally, the module should present. When integrating care services this module can be used to enhance patient-centeredness and inter-professional teamwork.

#### 3.2.22. Steering Committee

The steering committee, together with the project management team, will manage, control and organize the network. The strategic focus rests on defining responsibilities, planning of meetings, delegating tasks and coordinating WG’s. Additionally, the committee is responsible for providing transparent and comprehensive information to the partners in order to maintain their motivation. The committee is responsible for identifying and overcoming obstacles. The committee and the project management team should be composed of members from various disciplines. This module does not require contract-based collaboration or a legal framework. However, financial resources might be needed. Members of the committee should be willing to take responsibility and meet regularly. Additionally, the committee should communicate carefully and make sure that every member of the network receives the same information. Overall, the steering committee helps to designate roles and tasks within the network and foster inter-professional teamwork.

#### 3.2.23. Structured Patient School according to the Self-Management Concept

A structured patient school offers uniform training of PD patients to strengthen their participation and autonomy. Therefore, knowledge transfer interventions are combined with psycho-educative measures to support the patients’ competence in health-related self-management [28]. Self-management support has a top priority for patients in integrated care concepts [29]. In order to implement the module, various stakeholders, such as university hospitals, clinics and private partners, are required. A written cooperation agreement and financial resources will be needed. Partners cooperate based on needs and obligations within the agreement. The focus should rest on the application of appropriate education strategies, acquiring patients and conducting a continuous quality assessment. Externally, the module should be present. When it comes to integrating care, the school forms a part of the delivery system and helps to enhance patient-centeredness within the network.

#### 3.2.24. Virtual Visits

As it has been mentioned, the interlinking of inpatient and outpatient sectors is not common in Germany and experts often work independently from each other, which is especially problematic when implementing therapy options for complicated disease courses. One way to overcome this issue are virtual visits. They consist of electronic consultations of complex PD patients together with an expert at a university hospital and a resident neurologist. This provides the possibility to simplify communication. Additionally, the expert can advise the resident neurologist and the patient on advanced treatment options such as DBS. Within the German reimbursement scheme this intervention is funded, however, the university clinic doctor and the resident neurologist need to fulfil requirements according to the German eHealth act, which includes registration with a certified video consultation provider. This module facilitates the use of technology, contributes to the network delivery system within a network and improves patient-centeredness.

#### 3.2.25. Working Group

A WG is formed to process needs and orders derived from network meetings and develop targeted result oriented-solution proposals. Usually, partners with particular expertise and interest participate. Financial resources may be required for room rental or catering. Typically, WGs are formed in the case of a third-party assignment (e.g., expert opinion) or if there is a particular workload for the WG members. It may be practical to form WG during multi-disciplinary meetings, based on the voluntariness of the partners or on project-related tasks, such as the preparation of a press conference. Generally, WG collaborate on an ongoing basis when working on the assigned task, otherwise they are at rest. Externally, their work may be presented on the website, during lectures and mentioned in the minutes of multi-disciplinary meetings. They help to define clear roles and tasks within a network, foster inter-professional teamwork and improve the delivery system.

### 3.3. Expert Rating

The results of the expert rating can be derived from Table 1. Out of all categories, the highest ratings were given for the perceived patient benefit and the perceived specialists (Figure 3). The modules with the highest perceived patient benefit focused on treating and informing patients with the help of a structured patient school (mean = 4.56 ± 0.63), a regional utilities atlas (mean = 4.59 ± 0.51) and a Parkinson complex therapy (mean = 4.56 ± 0.73). Strengthening network communication through introducing Quickcards (mean = 4.40 ± 0.99), organizing multi-disciplinary plenary meetings (mean = 4.47 ± 0.72) and working groups (mean = 4.41 ± 0.71) were rated as interventions with the highest specialist benefit. Some modules seemed to be more time and resource consuming than others: especially when evaluating the network (additional workload: mean = 4.13 ± 0.83; resources required: mean = 4.07 ± 0.80) and performing outpatient video therapy sessions (expertise required: mean = 4.13 ± 0.96; resources required: mean = 4.13 ± 0.96). Organizing patient events were rated as modules with the lowest resource requirement (Info Café—resources required: 2.56 ± 0.73; expertise required: 2.75 ± 0.86; Summer Festival—expertise required: mean = 2.40 ± 0.91). Generally, interventions which did not involve patients or specialists directly were rated with lower perceived benefits, such as the Parkinson assistant (expected specialist benefit: mean = 2.38 ± 0.96), the Summer Festival (expected specialist benefit: mean = 2.47 ± 0.99) or the symposium (expected patient benefit: mean = 3.13 ± 0.92).

## 4. Discussion

This study gave an overview of German Parkinson networks and the modules that make up the networks. In addition, the modules were presented in terms of their relevance with regard to the design of an integrative care approach and their costs and benefits were evaluated by experts who work with them. This study offers an insight into the practical design of Parkinson networks that was previously not available.

Overall, the highest ratings were given for the perceived patient and specialist benefit, which indicates that Parkinson networks offer an added value to all stakeholders involved. Sustainable financing strategies are essential for ensuring the network’s long-term persistence [17]. However, in the beginning, other things may be prioritized, such as establishing ambitions and motivation among partners [19]. For example, senior leadership can be used as a tool to change stakeholders’ perception about perceived barriers to integrating care, such as a lack of resources [17]. Several of the described networks have been founded during the past years, which is why financing strategies may not be on the top of their agenda yet. Instead, tools which help to connect partners and exchange knowledge may be used to define, build and expand the network [19,20]. Defining and establishing the network can be a long process, depending on the size and readiness of the involved organizations but also on the regional context [17]. Building bridges across different care providers can be difficult but is of the utmost importance when setting up a network [17,19]. This has also been perceived by the German experts, which rated modules that enabled inter-professional exchange generally as beneficial for them.

By working in multi-disciplinary teams, not only patients, but the entire network benefits and tools, such as WG’s, plenary meetings and communication platforms are examples of how this can be achieved [20]. Other modules, such as the Parkinson Summer Festival or the Info-Café, may not be crucially important for the establishment or existence of a network. Still, they can function as a tool to include the patient’s perspective into the network. Putting the patient at the center of the network has been described as an essential feature of integrated care [17,19]. Considering patients as care partners, listening to their goals and inviting them to participate actively have been recommended as good practice for organizing care in PD [20]. Additionally, these modules may function as an engine to strengthen a patient’s commitment towards the network. For healthcare professionals, patient-centeredness may be a common-sense which can be used as a guiding value to bring different professions together and agree on joint actions [19].

Several modules focused on educating partners and training specialized staff for PD. Enabling interdisciplinary learning has been identified as a critical component of good care for PD [20]. Networks are suitable tools to enable learning and knowledge exchange [17,19]. Forming a network may reduce barriers for exchanging resources, including knowledge and expertise and ultimately increase the resources which are available to every single stakeholder within the network [14]. This has also been perceived by the German experts, who rated modules, which enabled knowledge exchange as beneficial for specialists and patients.

German Parkinson networks have not been established in a systematic way but rather depend on local initiatives [7]. Building care from a bottom-up approach has been pointed out as a successful approach when integrating care [14]. However, it has been recommended to systematically assess the cross-sectoral treatment quality for PD patients and foster activities which are coordinated on a national level [7]. Globally, the relevance of Parkinson networks has been recognized and in some European countries, such as the Netherlands and England, coordinated activities to build and improve such networks exist [12,21,30,31]. However, in large parts of the world, Parkinson networks are not present, which remains an issue that should be addressed in the future [12]. In Germany it has been acknowledged that results from other countries are only transferable to a limited extent since they focus on different priorities which may not be relevant for the German context [32]. Generally, the establishment of networks remains a great challenge in Germany due to inhomogeneous regional conditions and the strict regulation and separation of the individual sectors in the German health care system [32]. Overall, German authors agree that the healthcare delivery system for PD patients must change and that more Parkinson networks are needed in the future in order to supply all PD patients in Germany [13,16,22,33]. Establishing recommendations for building a Parkinson network that are generally agreed upon and specifically tailored to the German context may be a first step towards this direction.

For the future, Parkinson networks may not only provide a regional benefit but also may contribute to improving care for PD patients on a national level. Establishing and expanding networks may not only be a valuable contribution to the health of PD patients but also to the work of professionals and to the scientific community in Germany. PD is a very complex disease where care is mostly provided by hospitals and resident neurologists, which is why the use of existing integrated care networks for other chronic diseases (i.e., cancer) or the primary care network may not be sufficient. To the best of our knowledge, there are no such efforts for other neurodegenerative disease (i.e., Huntington disease). Therefore, to broaden the patient spectrum and also include other patient groups into the existing networks, is part of our vision. The modules that we have presented may be seen as a learning opportunity for existing and future Parkinson networks. While there is no doubt the appropriateness of each module depends on the context, state of development and perspective, this paper may be used as the first point for orientation.

At the same time, several questions remain open, which may be addressed in the future. What is the current level of integration in Germany? Is there something missing? How can we improve integration? Which modules are present in which network and how do the networks differ from each other? In order to answer these questions and foster network building, research should be facilitated and knowledge, as well as experiences, should be shared. Based on this study and the experiences, which were collected during the past years, the following recommendations for building a Parkinson network can be given:The purpose and focus of a network may vary across time and settings, but all aspects of integrated care should be covered in a network.When building or expanding a network module may be a valuable tool for orientation and priority may be given to modules with the best resource-benefit ratio.Fostering knowledge exchange and strengthening commitment are essential during each phase of development and sound project management should be present within each network.

## 5. Conclusions

Taking care of PD patients is complex and establishing networks, which integrate different care services, provides a huge benefit for patients and their families. Additionally, specialists can benefit as well, as it has been shown in this publication. Building a network is a complex procedure, which may differ from context to context. Goals and priorities can vary across time and different tools can be used to adapt to these needs. Using modules, which have been already implemented successfully, may be a good approach when establishing or expanding a network. In Germany, several Parkinson networks exist, however, delivering care based on an integrated care approach or within a network is not commonly present. Fostering the exchange of knowledge and expertise has been shown as a fundamental part across all German networks. Thus, future research may be used to improve understanding about Parkinson networks in Germany and examine how modules contribute to the integration of care in PD but also collect and share experiences and perceptions from other stakeholder groups. Overall, the German experience showed that Parkinson networks are a valuable improvement in treating PD patients.

## Figures and Tables

**Figure 1 jcm-09-02743-f001:**
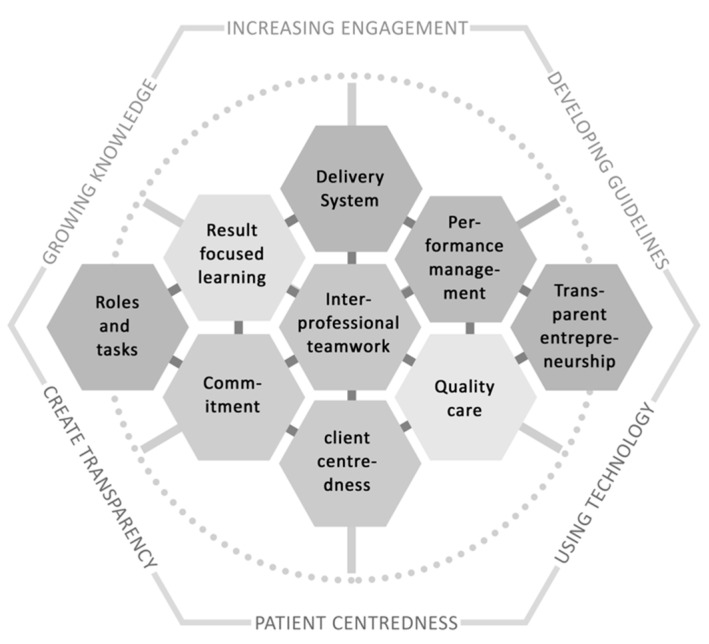
Core Aspects of Integrated Care for Parkinson’s Disease [18,19].

**Figure 2 jcm-09-02743-f002:**
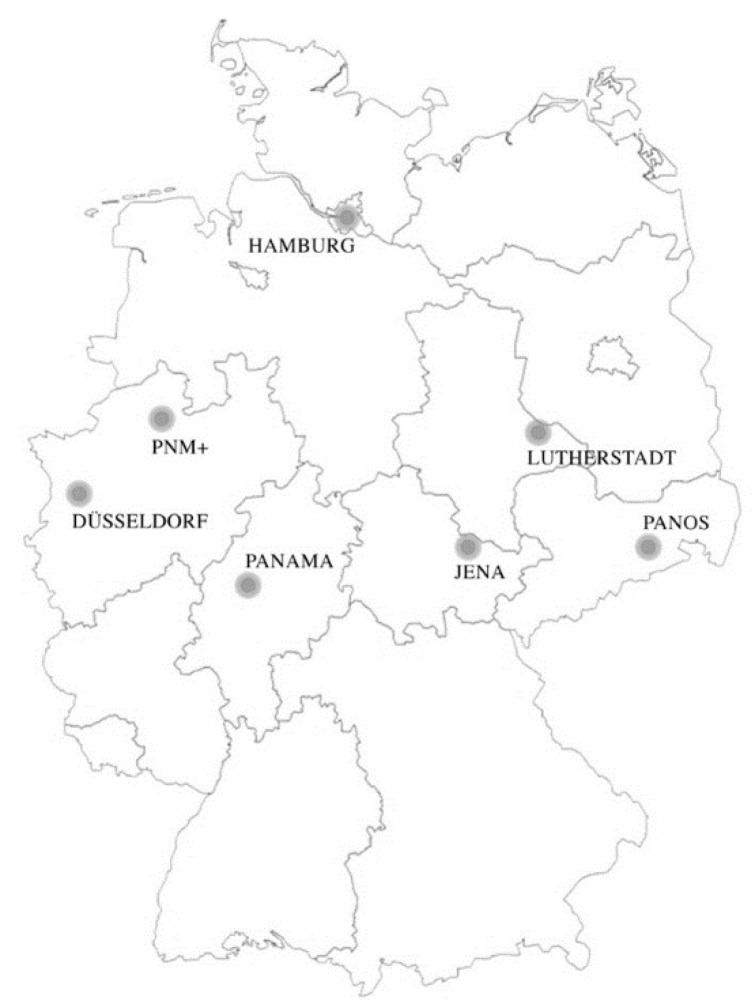
Parkinson Networks in Germany.

**Figure 3 jcm-09-02743-f003:**
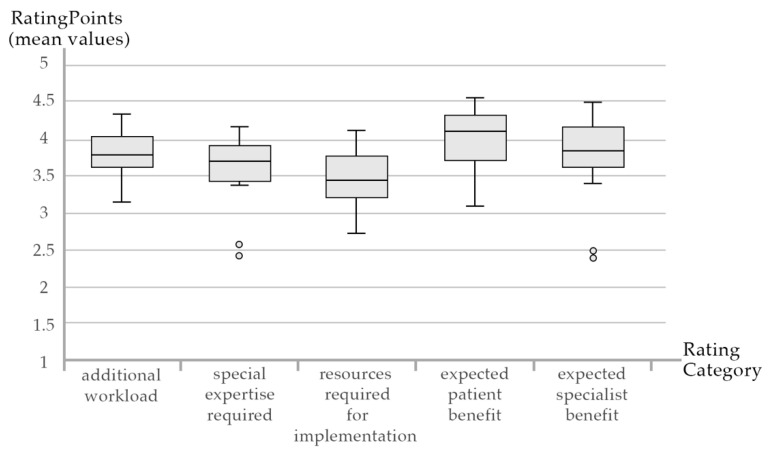
Box Plots for the Rating Categories and Rating Values (Modules: *n* = 25, Raters: *n* = 17).

**Table 1 jcm-09-02743-t001:** Expert Rating for German Parkinson Network Modules.

Module	Rating Criteria	Number of Responses	Mean (SD ^1^)	Visualized Results ^2^
**Deep Brain Stimulation Consultation**	Additional workload	*n* = 16	3.44 (± 0.89)	★ ★ ★ ☆ ☆
Special expertise required	4.06 (± 1.06)	★ ★ ★ ★ ☆
Resources required for implementation	3.19 (± 0.98)	★ ★ ★ ☆ ☆
Expected patient benefit	4.31 (± 1.01)	★ ★ ★ ★ ☆
Expected specialist benefit	3.94 (± 1.00)	★ ★ ★  ☆
**Evaluation**	Additional workload	*n* = 16	4.13 (± 0.83)	★ ★ ★ ★ ☆
Special expertise required	4.07 (± 1.10)	★ ★ ★ ★ ☆
Resources required for implementation	4.07 (± 0.80)	★ ★ ★ ★ ☆
Expected patient benefit	3.73 (± 1.16)	★ ★ ★  ☆
Expected specialist benefit	4.13 (± 0.83)	★ ★ ★ ★ ☆
**Financing Strategies**	Additional workload	*n* = 16	4.31 (± 0.60)	★ ★ ★ ★ ☆
Special expertise required	3.56 (± 0.81)	★ ★ ★  ☆
Resources required for implementation	3.94 (± 0.57)	★ ★ ★  ☆
Expected patient benefit	3.38 (± 1.20)	★ ★ ★ ☆ ☆
Expected specialist benefit	3.63 (± 1.09)	★ ★ ★  ☆
**Homepage**	Additional workload	*n* = 16	3.81 (± 0.83)	★ ★ ★  ☆
Special expertise required	3.38 (± 1.09)	★ ★ ★ ☆ ☆
Resources required for implementation	3.63 (± 0.81)	★ ★ ★  ☆
Expected patient benefit	3.88 (± 1.02)	★ ★ ★  ☆
Expected specialist benefit	3.81 (± 0.75)	★ ★ ★  ☆
**Multi-Disciplinary Plenary Meetings**	Additional workload	*n* = 17	3.88 (± 0.86)	★ ★ ★  ☆
Special expertise required	3.35 (± 1.06)	★ ★ ★ ☆ ☆
Resources required for implementation	3.47 (± 0.80)	★ ★ ★ ☆ ☆
Expected patient benefit	3.88 (± 0.60)	★ ★ ★  ☆
Expected specialist benefit	4.47 (± 0.72)	★ ★ ★ ★ ☆
**Multimodal Complex Treatment**	Additional workload	*n* = 16	3.75 (± 0.86)	★ ★ ★  ☆
Special expertise required	4.13 (± 0.96)	★ ★ ★ ★ ☆
Resources required for implementation	3.88 (± 0.89)	★ ★ ★  ☆
Expected patient benefit	4.56 (± 0.73)	★ ★ ★ ★ 
Expected specialist benefit	3.38 (± 1.09)	★ ★ ★ ☆ ☆
**Online Communication Platform**	Additional workload	*n* = 16	3.63 (± 1.20)	★ ★ ★  ☆
Special expertise required	3.44 (± 1.03)	★ ★ ★ ☆ ☆
Resources required for implementation	3.69 (± 0.87)	★ ★ ★  ☆
Expected patient benefit	3.94 (± 1.06)	★ ★ ★  ☆
Expected specialist benefit	4.38 (± 0.72)	★ ★ ★ ★ ☆
**Outpatient Video-Supported Therapy**	Additional workload	*n* = 16	4.00 (± 1.10)	★ ★ ★ ★ ☆
Special expertise required	4.13 (± 0.81)	★ ★ ★ ★ ☆
Resources required for implementation	4.13 (± 0.96)	★ ★ ★ ★ ☆
Expected patient benefit	4.44 (± 0.73)	★ ★ ★ ★ ☆
Expected specialist benefit	3.75 (± 1.06)	★ ★ ★  ☆
**Parkinson Academy**	Additional workload	*n* = 16	4.00 (± 0.65)	★ ★ ★ ★ ☆
Special expertise required	3.80 (± 0.94)	★ ★ ★  ☆
Resources required for implementation	347 (± 0.83)	★ ★ ★ ☆ ☆
Expected patient benefit	3.67 (± 0.82)	★ ★ ★  ☆
Expected specialist benefit	3.73 (± 0.96)	★ ★ ★  ☆
**Parkinson Assistant (PASS) Training**	Additional workload	*n* = 16	3.56 (± 1.09)	★ ★ ★  ☆
Special expertise required	3.38 (± 0.81)	★ ★ ★ ☆ ☆
Resources required for implementation	3.19 (± 0.91)	★ ★ ★ ☆ ☆
Expected patient benefit	4.13 (± 0.89)	★ ★ ★ ★ ☆
Expected specialist benefit	3.63 (± 1.15)	★ ★ ★  ☆
**Parkinson Board**	Additional workload	*n* = 16	3.38 (± 0.72)	★ ★ ★ ☆ ☆
Special expertise required	3.88 (± 1.02)	★ ★ ★  ☆
Resources required for implementation	2.75 (± 0.68)	★ ★  ☆ ☆
Expected patient benefit	4.00 (± 0.82)	★ ★ ★ ★ ☆
Expected specialist benefit	3.94 (± 1.06)	★ ★ ★  ☆
**Parkinson Nurse**	Additional workload	*n* = 16	3.75 (± 0.93)	★ ★ ★  ☆
Special expertise required	3.81 (± 0.83)	★ ★ ★  ☆
Resources required for implementation	3.56 (± 0.81)	★ ★ ★  ☆
Expected patient benefit	4.38 (± 0.89)	★ ★ ★ ★ ☆
Expected specialist benefit	3.50 (± 1.21)	★ ★ ★  ☆
**Parkinson Summer Festival**	Additional workload	*n* = 15	3.53 (± 0.83)	★ ★ ★  ☆
Special expertise required	2.40 (± 0.91)	★ ★ ☆ ☆ ☆
Resources required for implementation	3.47 (± 0.99)	★ ★ ★ ☆ ☆
Expected patient benefit	3.33 (± 1.11)	★ ★ ★ ☆ ☆
Expected specialist benefit	2.47 (± 0.99)	★ ★ ☆ ☆ ☆
**Parkinson Symposium**	Additional workload	*n* = 15	3.73 (± 0.80)	★ ★ ★  ☆
Special expertise required	3.73 (± 1.10)	★ ★ ★  ☆
Resources required for implementation	3.47 (± 0.64)	★ ★ ★ ☆ ☆
Expected patient benefit	3.13 (± 0.92)	★ ★ ★ ☆ ☆
Expected specialist benefit	4.27 (± 0.88)	★ ★ ★ ★ ☆
**Parkinson Info-Café**	Additional workload	*n* = 16	3.13 (± 0.81)	★ ★ ★ ☆ ☆
Special expertise required	2.56 (± 0.73)	★ ★  ☆ ☆
Resources required for implementation	2.75 (± 0.86)	★ ★  ☆ ☆
Expected patient benefit	4.13 (± 1.09)	★ ★ ★ ★ ☆
Expected specialist benefit	2.38 (± 0.96)	★ ★ ☆ ☆ ☆
**Quickcards**	Additional workload	*n* = 16	3.47 (± 0.74)	★ ★ ★ ☆ ☆
Special expertise required	3.67 (± 0.62)	★ ★ ★ ★  ☆
Resources required for implementation	3.40 (± 0.99)	★ ★ ★ ☆ ☆
Expected patient benefit	4.27 (± 1.03)	★ ★ ★ ★ ☆
Expected specialist benefit	4.40 (± 0.99)	★ ★ ★ ★ ☆
**Regional Demand Analysis and Target Formulation**	Additional workload	*n* = 16	3.69 (± 0.60)	★ ★ ★  ☆
Special expertise required	3.44 (± 0.81)	★ ★ ★ ☆ ☆
Resources required for implementation	3.38 (± 0.50)	★ ★ ★ ☆ ☆
Expected patient benefit	4.38 (± 0.62)	★ ★ ★ ★ ☆
Expected specialist benefit	4.13 (± 0.72)	★ ★ ★ ★ ☆
**Regional Public Relations**	Additional workload	*n* = 16	3.71 (± 0.69)	★ ★ ★  ☆
Special expertise required	3.53 (± 0.94)	★ ★ ★ ☆ ☆
Resources required for implementation	3.06 (± 0.83)	★ ★ ★ ☆ ☆
Expected patient benefit	3.94 (± 0.83)	★ ★ ★ ★ ☆
Expected specialist benefit	3.82 (± 0.88)	★ ★ ★  ☆
**Regional Utilities Atlas**	Additional workload	*n* = 17	4.00 (± 0.61)	★ ★ ★ ★ ☆
Special expertise required	3.35 (± 0.86)	★ ★ ★ ☆ ☆
Resources required for implementation	3.63 (± 0.62)	★ ★ ★  ☆
Expected patient benefit	4.59 (± 0.51)	★ ★ ★ ★ 
Expected specialist benefit	3.82 (± 0.81)	★ ★ ★  ☆
**Rehab**	Additional workload	*n* = 16	4.06 (± 0.85)	★ ★ ★ ★ ☆
Special expertise required	4.06 (± 1.24)	★ ★ ★ ★ ☆
Resources required for implementation	3.44 (± 1.03)	★ ★ ★ ☆ ☆
Expected patient benefit	4.19 (± 0.91)	★ ★ ★ ★ ☆
Expected specialist benefit	3.56 (± 1.15)	★ ★ ★  ☆
**Standardized Treatment Path**	Additional workload	*n* = 16	4.06 (± 0.68)	★ ★ ★ ★ ☆
Special expertise required	3.81 (± 0.98)	★ ★ ★  ☆
Resources required for implementation	3.69 (± 0.79)	★ ★ ★  ☆
Expected patient benefit	4.31 (± 0.79)	★ ★ ★ ★ ☆
Expected specialist benefit	4.00 (± 1.10)	★ ★ ★ ★ ☆
**Steering Committee/Project Management**	Additional workload	*n* = 17	3.81 (± 0.66)	★ ★ ★  ☆
Special expertise required	3.63 (± 1.02)	★ ★ ★  ☆
Resources required for implementation	3.44 (± 0.81)	★ ★ ★ ☆ ☆
Expected patient benefit	3.56 (± 1.15)	★ ★ ★  ☆
Expected specialist benefit	4.00 (± 0.89)	★ ★ ★ ★ ☆
**Structured Patient School according to the Self-management Concept**	Additional workload	*n* = 16	3.94 (± 0.85)	★ ★ ★  ☆
Special expertise required	3.81 (± 0.91)	★ ★ ★  ☆
Resources required for implementation	3.88 (± 0.81)	★ ★ ★  ☆
Expected patient benefit	4.56 (± 0.63)	★ ★ ★ ★ 
Expected specialist benefit	3.69 (± 1.01)	★ ★ ★  ☆
**Virtual Visits**	Additional workload	*n* = 16	3.81 (± 0.98)	★ ★ ★  ☆
Special expertise required	3.88 (± 1.02)	★ ★ ★  ☆
Resources required for implementation	3.88 (± 0.50)	★ ★ ★  ☆
Expected patient benefit	419 (± 0.75)	★ ★ ★ ★ ☆
Expected specialist benefit	3.56 (± 1.03)	★ ★ ★  ☆
**Working Groups**	Additional workload	*n* = 17	3.76 (± 0.83)	★ ★ ★  ☆
Special expertise required	3.47 (± 0.94)	★ ★ ★ ☆ ☆
Resources required for implementation	3.18 (± 0.64)	★ ★ ★ ☆ ☆
Expected patient benefit	3.76 (± 0.97)	★ ★ ★  ☆
Expected specialist benefit	4.41 (± 0.71)	★ ★ ★ ★ ☆

^1^ SD = Standard Deviation; ^2^

 = ≥0.5; ☆ = ≤0.5; ★ = 1.0; ★★ = 2.0; ★★★ = 3.0; ★★★★ = 4.0; ★★★★★ = 5.0.

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
