# Peer review of "Building a Parkinson-Network–Experiences from Germany"

_jcm, 2020, doi:10.3390/jcm9092743_

Round 1

Reviewer 1 Report

This is an interesting article on care networks in Parkinson’s disease in Germany. The paper is well written and provides concise and detailed information on the german expertise in this area. Patient centered approaches have great value and should be investigated in further studies. I have several comments/questions: 

Financing strategies: 

It would be interesting to get some some concrete information here. Did they co-operate with industry partners? Providing an estimate on the approximate financial costs f.ex. per year would be interesting.

Multimodal Complex Treatment:

Is this treatment program geared towards more advanced patients or very complicated cases ? If so, please add this in the text. 

Rehab: Please explain what rehabilitation centre are in this context? Are these hospitals with a rehab program or special nursing homes?

Discussion: It would be nice to discuss the situation of PD networks in other (european) countries and how it compares to Germany.

Introduction/Discussion: the authors should briefly state whether there are studies showing that such examples of care networks improve quality of life in PD patients. 

Author Response

Response the Reviewer

We would like to thank you for your contribution. We have addressed each point and described all changes that we have made in detail. Additionally, we highlighted all changes in the document. Please let us know if we can improve at any point!

Financing strategies: It would be interesting to get some concrete information here. Did they co-operate with industry partners? Providing an estimate on the approximate financial costs f.ex. per year would be interesting.

Response: We agree that this is an interesting point. However, the participant who filled out the template for financing strategies did not give concrete examples but rather described how a financing strategy is organized in general (you can access the full template in the supplementary material). For this paper, we think that this description is sufficient since financing strategies and annual costs can vary greatly depending on the region and the purpose of the network. From our conversations with network partners and the templates which have been filled out for the seven Parkinson networks it also became present that not all of them have established a financing strategy jet. This is a point that we would like to address more in detail in the future. However, we think that this is a valuable point and we inserted a short notice in the description.

Line 959 pp: Generally, financing strategies may differ depending on the regional context, the size of the network, the partners which are involved and the overall purpose and it has been recommended that these factors should be taken into account when establishing a strategy [10].

Multimodal Complex Treatment: Is this treatment program geared towards more advanced patients or very complicated cases? If so, please add this in the text. 

 Response: Indeed this treatment program is geared towards more advanced/ very complex cases and we have included this information in the text! Thank you!

Line 1023 pp. The Multimodal Complex Treatment is an in-patient treatment programme geared towards more advanced patients and very complicated cases.

Rehab: Please explain what rehabilitation centre are in this context? Are these hospitals with a rehab program or special nursing homes?

 Response: The expert is visiting a hospital with an outpatient rehab programme to educate the team there about advanced treatment methods in the statutory care setting. We have inserted this information in the text – Thank you for mentioning!

Line 1235 pp: This module compromises visits from an expert working in the statutory care sector (usually at a university hospital) at another hospital which offers a rehabilitation programme (outpatient care). During this visit, the expert educates the team about advanced treatment options (i.e. DBS) and gives advice on specific patient cases. The aim is to expand competence regarding advanced treatment options in external rehabilitation clinics and enhance patient safety and satisfaction. Similar to the DBS consultation, this module helps to connect the statutory and outpatient care sector

Discussion: It would be nice to discuss the situation of PD networks in other (European) countries and how it compares to Germany.

Response: We agree that this is a valuable perspective for the discussion section (also to highlight the need for more activities in other countries!). We have therefore included the following section:

Line 1659 pp: Globally, the relevance of Parkinson Networks has been recognized and in some European countries, such as the Netherlands and England, coordinated activities to build and improve such networks exist [11,20,29,30]. However, in large parts of the world Parkinson Networks are not present, which remains an issue that should be addressed in the future [11]. In Germany it has been acknowledged that results from other countries are only transferable to a limited extent since they focus on different priorities which may not be relevant for the German context [31]. Generally, the establishment of networks remains a great challenge in Germany due to inhomogeneous regional conditions and the strict regulation and separation of the individual sectors in the German health care system [31]. Overall, German authors agree that the healthcare delivery system for PD patients must change and that more Parkinson Networks are needed in the future in order to supply all PD patients in Germany [12,15,21,32]. Establishing recommendations for building a Parkinson Network that are generally agreed upon and specifically tailored to the German context may be a first step towards this direction.

Introduction/Discussion: the authors should briefly state whether there are studies showing that such examples of care networks improve quality of life in PD patients. 

Response: Thank you for this comment. We have cited studies in the introduction which showed that integrated care networks improve quality of life in PD patients. However, we added an example to underline the statement we made:

Line 56 pp: Integrating care services has been shown as an effective tool to meet these requirements [10,11] and as a method to enhance patients' quality of life and access to health care [10,12,13]. Integration can be achieved in various ways. For example, in 2018 a local initiative in Germany fostered the communication between several PD care providers which significantly improved the quality of life in PD patients compared to the standard neurological practice [10]. Overall, the establishment of Parkinson Networks has been shown as a useful tool to achieve the integration of care services [11,14]. Parkinson Networks can be understood as a merger of several units that share and exchange data, as well as resources among each other [15]. They often aim to improve patient care across different sectors, avoid unnecessary hospitalizations and reduce costs. However, these networks can also be used to enable faster and more efficient PD diagnosis, optimize patient treatment and create better working conditions for healthcare providers [9,15].

Response to the other Reviewers

In order to give you a better overview about the points of the other reviewers we have listed all additional changes here:

  • English Editing
  • Reducing the word count
  • Addressing access to the network and coverage of Germany
  • Difference to care for other neurodegenerative diseases
  • Consistently using “patient-centeredness”
  • Improve method section
  • Reorganize result section and table 1
  • Study summary at the beginning of the discussion
  • Statistics about PD in Germany
  • Discuss the current care situation (can primary care network or other integrated care networks be used for PD)
  • Point out the need for uniform guidelines

Reviewer 2 Report

This is an outstanding description of the German Parkinson Network, which is also relevant for other health systems.

The overall suggestion would be to reduce the word No. by 20-30% to improve the readibility, the text is a bit verbous in some parts.

Regarding the content:

-Which actions guarantee that PD patients reach the Network, i.e. given that there are only 7 Centers and that they do not cover all German regions, how to be sure that gp and General neurologists indeed refer their PD patients to the dedicated events of the Network?

-In which way does th eParkinson Network differ from others f.i. for atypical Parkinsonian disorders or other neurodegenerative diseases like Huntington Disease?

-Virtual visits: why are both an expert and a resident neurologist needed? Who is on the other end? The Patient alone? This is unclear.

Minors:

  • Autonomous>autonomic
  • the authors sometimes use Patient-centeredness, sometimes client-centeredness
  • Orientated>oriented

Author Response

Response the Reviewer

We would like to thank you for your contribution. We have addressed each point and described all changes that we have made in detail. Additionally, we highlighted all changes in the document. Please let us know if we can improve at any point!

The overall suggestion would be to reduce the word No. by 20-30% to improve the readability, the text is a bit verbous in some parts.

Response: We have received feedback from other reviewers who wished more information in several chapters (i.e. the introduction and the discussion) so it was difficult to find a compromise. However, we agree that some sections can be shortened so we went through the text again and tried to reduce the word No. as much as possible.

After including all relevant aspects and comments we counted 11578 words and we reduced the word count to 10804 words. We have attached the revised document including all changes that we have made.

Regarding the content:

Which actions guarantee that PD patients reach the Network, i.e. given that there are only 7 Centers and that they do not cover all German regions, how to be sure that GP and General neurologists indeed refer their PD patients to the dedicated events of the Network?

Response: This is an issue which has to be addressed in Germany and which is why we have decided to provide an overview of all networks in Germany. In Germany, healthcare is organized on a regional level, which is why some regions do not have a Parkinson network and others do. Currently, it´s up to the networks to connect with GP´s and resident neurologists which is very time and resource consuming. It is not guaranteed that patients are referred to the network by their GP or resident neurologist. However, we believe that there are things that can be done to motivate GP´s/ resident neurologists to refer their patients to the network – for example by including them actively in the network (i.e. at plenary meetings). From our perspective, more networks and modules should be established. We decided to explain the current situation in Germany more in detail and included the following information in the text to better highlight the need for more integrated care services related to Parkinson´s disease. We have inserted the following explanation in the text:

Line 94-pp (Introduction): Furthermore, healthcare in Germany is organized on a regional level and Parkinson Networks heavily rely on local initiatives [12,21]. At the moment, neither all PD patients have access to Parkinson Network, nor it is guaranteed that all PD patients within the reach of a network are referred to it by their primary care provider [5]. (…) Today, the establishment of a Parkinson Network which interlinks important care providers, such as resident neurologists and experts at a hospital, is time and resource consuming. The networks have to come up with own strategies to connect with local care providers and patients and currently, there are only little practice-based examples how this could be achieved [8]. This paper presents an overview of interventions (so-called ‘modules‘) which have been used to build these networks and, for example, helped to ensure that resident neurologists indeed refer their PD patients to the dedicated events of a network.

Line 310 pp (Results): What becomes present is that not all German regions have implemented a Parkinson Network.

Line 1260 pp (Regional Demand Analysis): Analyzing regional demand may help to detect current issues, such as a lack of interface collaboration or barriers for patients to enter the network (i.e. finding reasons for why resident neurologist/ general practitioners are not referring their patients to the network).

Line 1094 pp (Multi-disciplinary Plenary Meetings): Plenary meetings make it possible for the partners to get to know each other. Additionally, these meetings provide a platform for regular exchange, voting about results, making suggestions to other project groups and collecting new ideas. Thus, these meetings can help to build trust and documenting progress towards overall goals. Additionally, these meetings may be used to enhance the motivation of general practitioners and resident neurologists to refer their PD patients to the dedicated events of a Network

In which way does the Parkinson Network differ from others f.i. for atypical Parkinsonian disorders or other neurodegenerative diseases like Huntington Disease?

Response: Most of the Parkinson Networks also include patients with atypical Parkinsonian disorders. There are, to the best of our knowledge, no networks present in Germany for other neurodegenerative diseases, such as Huntington Disease which is why we can´t make a comparison here. For the future it is our envision to broaden the patient spectrum and also include other patient groups since we believe that also other patient groups would benefit from integrated care networks. We have included the above-mentioned information in the text.

Line 76 pp (Introduction): In Germany there are no well-established multi-disciplinary care structures for neurological diseases in general. However, when it comes to PD, care concepts have developed dynamically within the past years and today several integrated care networks for PD patients (usually also including patients with atypical Parkinsonian disorders) exist in Germany, for example, in Münster, Dresden and Marburg.

Line 1786 pp (Discussion): PD is a very complex disease where care is mostly provided by hospitals and resident neurologists, which is why the use of existing integrated care networks for other chronic diseases (i.e. cancer) or the primary care network may not be sufficient. To the best of our knowledge, there are no such efforts for other neurodegenerative disease (i.e. Huntington disease). Therefore, we envision to broaden the patient spectrum and also include other patient groups into the existing networks.

Virtual visits: why are both an expert and a resident neurologist needed? Who is on the other end? The Patient alone? This is unclear.

Response: These two parties often deliver different care services which are both relevant for PD patients. Currently, there is no stipulation in Germany who will initiate and continue treatment for PD patients. In particular, planning and implementation of therapy options for complicated and advanced disease courses reach their limits without an optimal setting. A survey of German Parkinson's patients in 2018 showed that local neurologists are the first point of contact for the patients. Another important care provider are experts at a hospital, which provide advanced treatment options and statutory care. However, there is no regular interlinking of the inpatient and outpatient sectors and in some cases, this can lead to non-coordinated approach to diagnostics and therapy. Having both healthcare professionals present during a meeting can help to overcome this barrier. We recognize that this is not intuitive and addressed this in the introduction. Additionally, we inserted a small explanatory section in chapter 3.1.25.

Line 86 pp (Introduction) A survey of German PD patients in 2018 showed that the first point of contact usually is in the outpatient sector with resident neurologists being the main care provider to the patients (85.2%, n=1179) [12] (p.708). Most of the PD patients consult a resident Neurologists or Psychiatrists (90.8%; n=1156), a hospital (21.2%; n=255) and a specialized Parkinson Clinic (17.7%; n=225) at least once per year [12] (p.708). Experts at hospitals and clinics are important stakeholders in the statutory care sector since they provide advanced treatment methods (i.e. deep brain stimulation (DBS)) [12]. Currently, there is no regular mechanisms to interlink statutory and outpatient care. This can lead a non-coordinated approach to diagnostics and therapy [12].

Line 1410 pp (Virtual Visits): As it has been mentioned, the interlinking of inpatient and outpatient sectors is not common in Germany and experts often work independently from each other, which is especially problematic when implementing therapy options for complicated disease courses. One way to overcome this issue are virtual visits. They consist of electronic consultations of complex PD patients together with an expert at a university hospital and a resident neurologist. This provides the possibility to simplify communication. Additionally, the expert can advise the resident neurologist and the patient on advanced treatment options such as DBS.

Minors:

Autonomous>autonomic

Response: this has been corrected in line 43 – thank you!

the authors sometimes use Patient-centeredness, sometimes client-centeredness

Response: Minkman defined one aspect of good integrated care as having a network that is centered around the client (which is in our case equivalent to the patient). However, we recognize that this might have caused some confusion and thus decided to use “patient-centeredness” consistently across the text since this terminology has also been used in other publications relating to integrated care for PD.

We have introduced the term “patient centeredness” in the method section (line 205)

… and the term ‘client-centeredness’ has been replaced by the term ‘patient-centeredness’…

and used it then consistently throughout the text

Orientated>oriented

Response: this has been corrected in line 340, 1618 – thank you

Response to the other Reviewers

In order to give you a better overview about the points of the other reviewers we have listed all additional changes here:

  • English Editing
  • Consistently using “patient-centeredness”
  • Improve method section
  • Reorganize result section and table 1
  • Study summary at the beginning of the discussion
  • Comments regarding financing strategies, multimodal complex treatment and Rehab
  • Discussion of PD networks in other EU countries
  • Studies showing that networks improve QoL in PD patients
  • Statistics about PD in Germany
  • Discuss the current care situation (can primary care network or other integrated care networks be used for PD)
  • Point out the need for uniform guidelines

Reviewer 3 Report

This article covers an interesting and important topic that many can learn from. It will certainly be of interest to the readership. The authors did a nice job describing "integrated care" in the introduction and laying a foundation for the work described in the manuscript. The following are provided as general and specific comments for improvement:

  1. Overall, the English language and style require moderate changes. Care should be taken to edit the English language and style prior to resubmitting a new draft.
  2. The authors should take care to use consistent terminology. It is unclear whether modules = networks or modules = categories or something else. Please clarify terminology and use it consistently throughout the manuscript. In general, a term like “categories” is easier to interpret than is the term “modules”. However, if the term “modules” is selected, please ensure consistency throughout the text (especially in the Methods section).

  3. How was the information collected for this study? Web survey? Phone interview? Was there a specific point of contact for each network? Additional specific information is required in the methods section to describe how the information was collected, from whom, and over what period of time. How was data analysis conducted? Was qualitative analysis used? This section must be significantly expanded to outline key methodological steps in the study.

  4. It is not clear how the sub-sections of the network modules section (3.1) were derived. Were these themes identified from the questionnaire/information collection? If so, this should be clearly delineated. There should be a more clear introduction to this section to answer this question. How were the sections organized? Is there any rationale for organizing them in the current order? If not, how can they most logically be organized (most to least important, most to least benefit, alphabetical, etc.)?

  5. In Table 1, it would be easier to interpret half/partial black stars rather than grey stars for a non-full star rating. It would also help to organize this table in the same order as the section 3.1 sub-sections for consistency.
  6. I would recommend placing the network descriptions first before the descriptions of the modules/categories. It would help to have a better understanding of each network before delving into the information obtained from these networks.
  7. The discussion brings up additional information that should be in the results section. Lines 558-560 indicate that not all categories were present in all networks. This should be brought forth in the results, perhaps through the use of a table listing all 25 categories and a count (%) of how many networks utilized the strategy. This information would also help organize section 3.1 and Table 1.
  8. The discussion should begin with a summary paragraph of the study before jumping into discussing and interpreting the results.

  9. Line 38 should read “autonomic”, not “autonomous”.

Author Response

Response the Reviewer

We would like to thank you for your contribution. We have addressed each point and described all changes that we have made in detail. Additionally, we highlighted all changes in the document. Please let us know if we can improve at any point!

Overall, the English language and style require moderate changes. Care should be taken to edit the English language and style prior to resubmitting a new draft.

Response: We have edited the entire text again and had it checked by a native speaker to ensure that the grammar is correct. We have attached the revised document including all changes that we have made. If there are still any things we should change please let us know how we can improve!

The authors should take care to use consistent terminology. It is unclear whether modules = networks or modules = categories or something else. Please clarify terminology and use it consistently throughout the manuscript. In general, a term like “categories” is easier to interpret than is the term “modules”. However, if the term “modules” is selected, please ensure consistency throughout the text (especially in the Methods section).

Response: Thank you for this suggestion! This paper has a two-folded aim: First, it describes all Parkinson Networks in Germany and second, it shows the building blocks of which these networks are composed of. When it comes to healthcare system research – the terminology “modules” is used to describe “building blocks” and “linking structures” of a network. This term is often used in the literature which is why we decided to choose this terminology as well. We have slightly reformulated our study aim and inserted this explanation in the introduced to ensure a higher transparency. Additionally, we used the term “module” consistently across the text.

Line 100 pp: Today, the establishment of a Parkinson Network which interlinks important care providers, such as resident neurologists and experts at a hospital, is time and resource consuming. The networks have to come up with own strategies to connect with local care providers and patients and currently, there are only little practice-based examples how this could be achieved [8]. This paper presents an overview of interventions (so-called ‘modules‘) which have been used to build these networks and, for example, helped to ensure that resident neurologists indeed refer their PD patients to the dedicated events of a network.

Overall, this paper has a two-folded aim: First, it aims to present all Parkinson Networks and modules which are present in Germany. Second, it aims to highlight the contribution of each module to the establishment of an integrating care network and provide an expert judgement on their costs and benefits. Finally, recommendations for building a network based on German experiences will be given.

How was the information collected for this study? Web survey? Phone interview? Was there a specific point of contact for each network? Additional specific information is required in the methods section to describe how the information was collected, from whom, and over what period of time. How was data analysis conducted? Was qualitative analysis used? This section must be significantly expanded to outline key methodological steps in the study.

Response: Thank you for addressing these points. We have addressed all of the above-mentioned points in our methods sections. If you wish more information, please let us know how we could provide you with them!

Line 175 pp: In order to fulfill the first purpose of the investigation (to present an overview of all Parkinson Network and their modules) information about existing modules and networks was collected by using a Microsoft Word template.  The template was developed by the DPG working group 'Networks & Care' in which the authors participate to ensure a structured collection process. The template asked about the goal, strategic focus, participating parties, contractual cooperation partners, the legal framework, funding mechanisms, the structure and frequency of cooperation and external. It was used to collect information from October 2019 to December 2019 about networks and modules among experts from all over Germany. To ensure that all existing networks and modules have been compiled, the heads of each Parkinson Network were identified via the DPG registry and invited by mail to contribute to the study. The authors had access to the DPG registry since they are members of this association as well. Additionally, an invitation in form of an electronic newsletter was sent to the German PD expert community to participate in the study. Again, these experts were identified with the help of the DPG registry and included doctors and scientists focusing on care supply for PD patients. In both cases the template was attached to the invitation and participants were asked to describe a Parkinson Network in which they participate and/ or as many network modules as they wish by filling out the template. The participants were instructed to fill out one template for each module or network and sent them back to the investigators by mail. In order to provide an overview of the costs and benefits of each module, an anonymous web-based survey was conducted in 2020 from June 29th to July 11th among the participants from the template collection. They were invited to participate in the survey by email, which also included the link to access the survey. The experts rated each identified module on a five-point Likert-scale (1=none, 5=very much) according to the additional workload caused by its implementation, the expertise and amount resources required and the expected benefit for patients and specialists. Using a five-point Likert-scale as tool to retrieve expert opinions is a widely used method in integrated care research [22–24]. The rating criteria have been selected based on the recommendations for the organization of multi-disciplinary PD care teams [19]. The survey was analyzed by using Microsoft Excel to calculate the average rating points, as well as the standard deviation, for each category and each module. The results have been converted into a visual scheme to provide a better overview. Additionally, the modules have been grouped into the nine categories of the DMIC to facilitate a better overview of their part within an integrated care network. For the purpose of this study, PD specific criteria for care have been added to the model and the term ‘client-centeredness’ has been replaced by the term ‘patient-centeredness’ [19]. An overview of the categories can be derived from figure 1.

It is not clear how the sub-sections of the network modules section (3.1) were derived. Were these themes identified from the questionnaire/information collection? If so, this should be clearly delineated. There should be a more clear introduction to this section to answer this question. How were the sections organized? Is there any rationale for organizing them in the current order? If not, how can they most logically be organized (most to least important, most to least benefit, alphabetical, etc.)?

Response: the network modules were collected by sending a blanc template to various experts working in a Parkinson Network. The experts filled out the template for as many modules as they wished and sended them back to the authors by e-mail. We have included this information in the methods section and inserted a short introduction before presenting the modules. What would have been interesting is to group the modules according to their “cost-benefit” ratio. However, to the best of our knowledge, no comparable work exists so far which could have been used as guidance for the authors when calculating such a ratio. Thus, we have decided to sort the modules according to their alphabetical order and we are planning to conduct such a cost-benefit analysis in the near future.

Line 716: Next, the network modules are presented which have been collected using the template among PD experts working in one of the seven previously outlined Parkinson Networks. All templates that were handed in can be accessed in the supplementary material. The modules were organized in an alphabetical order.

In Table 1, it would be easier to interpret half/partial black stars rather than grey stars for a non-full star rating. It would also help to organize this table in the same order as the section 3.1 sub-sections for consistency.

Response: We have discussed this issue as well and we agree that half starts are better! We have been in contact with the JCM editor board since we did not know if it´s allowed to include a .png or .jpeg file in a table (MS Word does not allow to format a “half filled” star when using the pictogram feature). We received positive feedback from the editor board, so we changed the stars! Also, we reorganized the table and included partial filled stars.

These changes can be seen from line 1574 onward.

I would recommend placing the network descriptions first before the descriptions of the modules/categories. It would help to have a better understanding of each network before delving into the information obtained from these networks.

Response: we agree and changed the order! Thank you for this suggestion.

These changes can be seen from line 308 onward

The discussion brings up additional information that should be in the results section. Lines 558-560 indicate that not all categories were present in all networks. This should be brought forth in the results, perhaps through the use of a table listing all 25 categories and a count (%) of how many networks utilized the strategy. This information would also help organize section 3.1 and Table 1.

Response: We agree that this statement is not appropriate in the discussion section. Thank you for mentioning! The only data that is currently available to the authors is which participant handed in which module and their affiliation (which is of course an indicator for which module is present in which network). However, several participants wished to stay anonymous which is why we have not included information about the presence/ absence of modules in each network since this might be an indicator who contributed to the study. We agree that it would be an interesting investigation for the future to analyze each network in detail and we are planning to do so! We have reformulated our discussion section and we included an additional research question at the end of this section

Line 1764 pp: At the same time, several questions remain open, which may be addressed in the future. What is the current level of integration in Germany? Is there something missing? How can we improve integration? Which modules are present in which network and how do the networks differ from each other?

The discussion should begin with a summary paragraph of the study before jumping into discussing and interpreting the results.

Response: Indeed this gives the reader a better introduction to the discussion! We have included the following paragraph:

Line 1611 pp: This study gave an overview of German Parkinson Networks and the modules that make up the networks. In addition, the modules were presented in terms of their relevance with regard to the design of an integrative care approach and their costs and benefits were evaluated by experts who work with these modules. This study offers an insight into the practical design of Parkinson Networks that was previously not available.

Line 38 should read “autonomic”, not “autonomous”.

Response: this has been corrected in line 43 – thank you!

Response to the other Reviewers

In order to give you a better overview about the points of the other reviewers we have listed all additional changes here:

  • Reducing the word count
  • Addressing access to the network and coverage of Germany
  • Difference to care for other neurodegenerative diseases
  • Comments regarding financing strategies, multimodal complex treatment, virtual visits and Rehab
  • Discussion of PD networks in other EU countries
  • Studies showing that networks improve QoL in PD patients
  • Statistics about PD in Germany
  • Discuss the current care situation (can primary care network or other integrated care networks be used for PD)
  • Point out the need for uniform guidelines

Reviewer 4 Report

The authors describe the work on This paper is not a research paper by a strict definition, but rather an interesting recommendatin for standards of network care for patients with PD in Germany. The paper could be further improved by the following minor additions: 

 1. Add Statistics about PD in Germany including visits to different professions.

2. Discuss how and to which degree it is preferable to use existing care networks for other chronic diseases when involving primary care and the possible regional differences.

3. Point out even clearer the need for guidelines that are generelly agreed upon for the build-up of networks.

Author Response

We would like to thank you for your contribution. We have addressed each point and described all changes that we have made in detail. Additionally, we highlighted all changes in the document. Please let us know if we can improve at any point!

Add Statistics about PD in Germany including visits to different professions.

Response: Including numbers gives the reader a better impression of PD in Germany – thank you for this suggestion. We have included numbers about the prevalence of PD in Germany, and we included numbers on the visits of different professions. However, we must acknowledge that the evidence is very limited and there is currently only one study describing what professions are consulted by PD patients in Germany.

Line 36 pp According to data from the Federal Statistical Office, were 11,190 deaths related to the primary Parkinson's syndrome in 2018 [3]. The Germany society is aging and age-related diseases like PD will continue to increase Dorsey et al. estimates for the year 2030 that the prevalence of PD in Germany will increase by 36% (based on the starting year 2005) up to 150,000 cases [4]. However, this study only included prevalence´s from people aged 65 years and older and was based on a small sample, which is why the prevalence may be underestimated [5].

Line 86 pp: . A survey of German PD patients in 2018 showed that the first point of contact usually is in the outpatient sector with resident neurologists being the main care provider to the patients (85.2%, n=1179) [12] (p.708). Most of the PD patients consult a resident Neurologists or Psychiatrists (90.8%; n=1156), a hospital (21.2%; n=255) and a specialized Parkinson Clinic (17.7%; n=225) at least once per year [12] (p.708).

Discuss how and to which degree it is preferable to use existing care networks for other chronic diseases when involving primary care and the possible regional differences.

Response: This is a valuable input and we discussed it from two perspectives!

#1 using existing integrated care networks for other diseases or the national primary care network in order to supply PD patients: Integrated care networks are present for other diseases in Germany and they may have some similarities with Parkinson Networks (i.e. a communication platform). However, PD is such a complex disease that specific ways have to be found in order to supply these patients and these structures are not comparable with other integrated care networks, which is why it seems not preferable to use existing care networks. Also, primary care networks seem to be not relevant in the case of PD since the primary care supply for PD patients in Germany is done by hospitals and resident neurologists instead of general practitioners. 

#2 using Parkinson Networks to supply patients with other neurological diseases: Discussing this aspect was also recommended by another reviewer. We argue that this should be the long-term goal for the German Parkinson Networks, and we have included this envision in the outlook.

Line 1681 pp:  PD is a very complex disease where care is mostly provided by hospitals and resident neurologists, which is why the use of existing integrated care networks for other chronic diseases (i.e. cancer) or the primary care network may not be sufficient. To the best of our knowledge, there are no such efforts for other neurodegenerative disease (i.e. Huntington disease). Therefore, we envision to broaden the patient spectrum and also include other patient groups into the existing networks.

Point out even clearer the need for guidelines that are generally agreed upon for the build-up of networks.

Response: We agree that this is a key point in this publication! Other reviewers wished for a more international perspective (what are other countries currently doing) and we combined this outlook with an outlook on the future of Parkinson Networks in Germany and the importance of generally agreed upon recommendations.

Line 1659 pp: Globally, the relevance of Parkinson Networks has been recognized and in some European countries, such as the Netherlands and England, coordinated activities to build and improve such networks exist [11,20,29,30]. However, in large parts of the world Parkinson Networks are not present, which remains an issue that should be addressed in the future [11]. In Germany it has been acknowledged that results from other countries are only transferable to a limited extent since they focus on different priorities which may not be relevant for the German context [31]. Generally, the establishment of networks remains a great challenge in Germany due to inhomogeneous regional conditions and the strict regulation and separation of the individual sectors in the German health care system [31]. Overall, German authors agree that the healthcare delivery system for PD patients must change and that more Parkinson Networks are needed in the future in order to supply all PD patients in Germany [12,15,21,32]. Establishing recommendations for building a Parkinson Network that are generally agreed upon and specifically tailored to the German context may be a first step towards this direction.

Round 2

Reviewer 3 Report

Thank you for the improvements made to your manuscript. The authors have addressed all reviewer comments and the article is now suitable for publication.